# Intravital calcium imaging of meningeal macrophages reveals niche-specific dynamics and aberrant responses to brain hyperexcitability

**Simone Carneiro-Nascimento, Chao Wei, Anna Gutterman, Dan Levy***

Department of Anesthesia, Critical Care and Pain Medicine, Beth Israel Deaconess Medical Center, Harvard Medical School, Boston, United States

## eLife Assessment

This study provides **important** insights into how immune cells in the brain's protective layers behave under normal and disease-like conditions, revealing location-specific activity patterns that may shape inflammation and disorders such as migraine. The evidence is **compelling** and supported by advanced imaging approaches and rigorous analyses, although some conceptual and interpretational limitations temper the mechanistic depth. Overall, the work will be of broad interest and represents an invaluable contribution to the growing field linking immune and nervous system function.

*For correspondence:
dlevy1@bidmc.harvard.edu

Competing interest: The authors declare that no competing interests exist.

**Abstract** The meninges, which envelop and protect the brain, host a dense network of resident macrophages with diverse roles in regulating homeostasis and neuroinflammation. Despite their importance, we have a limited understanding of their behavior in vivo. Many dynamic cellular functions of macrophages involve intracellular $Ca^{2+}$ signaling. However, virtually nothing is known about the spatiotemporal $Ca^{2+}$ dynamics of meningeal macrophages in vivo. We developed a chronic intravital two-photon imaging approach and related computational analysis tools to interrogate meningeal macrophage $Ca^{2+}$ dynamics, at subcellular resolution, in a novel Pf4-Cre:Ai162 conditional GCaMP6s reporter mouse model. Using imaging in awake mice, we characterized $Ca^{2+}$ activity in meningeal macrophages at steady state and in response to cortical spreading depolarization (CSD), an aberrant pro-inflammatory brain hyperexcitability event implicated in migraine, traumatic brain injury, and stroke. In homeostatic meninges, macrophages in the dural perivascular niche exhibited several $Ca^{2+}$ dynamic features, including event duration and signal frequency spectrum, distinct from those localized to the interstitial, non-perivascular niche. Simultaneous tracking of macrophage $Ca^{2+}$ dynamics and local vasomotion revealed a subset of dural perivascular macrophages whose activity was coupled to locomotion-driven diameter fluctuations of their associated vessels. Most perivascular and non-perivascular meningeal macrophages displayed propagating intracellular $Ca^{2+}$ activity and synchronized intercellular $Ca^{2+}$ elevations, potentially driven by extrinsic factors. In response to CSD, the majority of perivascular and non-perivascular meningeal macrophages showed a persistent decrease in $Ca^{2+}$ activity, while a smaller subset displayed $Ca^{2+}$ elevations. Mechanistically, calcitonin gene-related peptide receptor signaling mediated the increase but not the decrease in CSD-mediated $Ca^{2+}$ signaling. Collectively, our results highlight a previously unknown diversity of $Ca^{2+}$ dynamics in meningeal macrophages at steady state and in response to an aberrant brain hyperexcitability event linked to neuroinflammation.

## Introduction

Resident macrophages are key myeloid immunocytes that play an important role in innate immune surveillance and defense across various peripheral tissues and organs (*Okabe and Medzhitov, 2016*; *Guilliams et al., 2020*; *Mass et al., 2023*). The central nervous system also harbors a large subset of parenchymal macrophages, known as microglia, and several distinct subsets of macrophages localized to the brain's border tissues, including the choroid plexus, perivascular spaces, and the meningeal compartments that cover, protect, and support the brain (*Kierdorf et al., 2019*; *Rustenhoven et al., 2021*; *Drieu et al., 2022*; *Masuda et al., 2022*; *Amann et al., 2024*). Macrophages are the predominant immune cell type within the brain meninges, and recent studies have demonstrated their diverse ontogeny, transcriptomic profiles, and immune functions at steady state (*Rustenhoven et al., 2021*; *Amann et al., 2024*; *Mrdjen et al., 2018*; *Van Hove et al., 2019*; *Smyth et al., 2024*; *Vara-Pérez and Movahedi, 2025*) and in several neuropathological conditions (*Amann et al., 2024*; *Rua et al., 2019*; *Rebejac et al., 2022*; *De Vlaminck et al., 2022*; *Pinho-Ribeiro et al., 2023*).

Cytoplasmic calcium ($Ca^{2+}$) signaling underlies a wide variety of cellular homeostatic and inflammatory processes in macrophages (*Desai and Leitinger, 2014*; *Zumerle et al., 2019*; *Seegren et al., 2020*; *Nascimento Da Conceicao et al., 2021*; *Taghdiri et al., 2021*; *Mehari et al., 2022*; *Seegren et al., 2023*). In addition to intracellular $Ca^{2+}$ elevation, distinct spatiotemporal dynamics—including oscillation patterns, intracellular propagations, and intercellular synchronization of $Ca^{2+}$ signals—may regulate different macrophage functions during steady state and pathophysiology (*Nascimento Da Conceicao et al., 2021*; *Taghdiri et al., 2021*; *Mehari et al., 2022*; *Seegren et al., 2023*; *Vaeth et al., 2015*; *Schappe et al., 2022*). Despite our increased understanding of the diverse molecular signatures and contributions of meningeal macrophages to homeostasis and neuroinflammation, virtually nothing is known about their $Ca^{2+}$ signaling heterogeneity in both healthy and diseased states.

Here, we comprehensively characterized the $Ca^{2+}$ dynamics of individual macrophages localized to the brain meninges by combining intravital two-photon $Ca^{2+}$ imaging in a novel reporter mouse line, in which the $Ca^{2+}$ reporter GCaMP6s is expressed in platelet factor 4 ($Pf4^+$) meningeal macrophages, with an event-based signaling analysis pipeline. Our data reveal several distinct spatiotemporal $Ca^{2+}$ dynamic features in perivascular versus interstitial non-perivascular meningeal macrophages, including a unique coupling between the $Ca^{2+}$ signals of dural perivascular macrophages and behaviorally driven vasomotion of their associated dural vessels at steady state. Furthermore, our data uncover both increases and decreases in $Ca^{2+}$ activity in distinct subsets of meningeal macrophages in response to cortical spreading depolarization (CSD), a pathophysiological brain hyperexcitability event linked to headache pain and neuroinflammation (*Kaya et al., 2025*) in migraine, traumatic brain injury, and stroke (*Harriott and Ayata, 2025*). Mechanistically, our data suggest that calcitonin gene-related peptide (CGRP) receptor signaling mediates CSD-evoked macrophage $Ca^{2+}$ elevation and related brain-to-meninges neuroimmune signaling pathway, potentially involving CGRP released from sensitized meningeal sensory neurons acting on macrophage CGRP receptors (*Pinho-Ribeiro et al., 2023*; *Levy and Moskowitz, 2023*; *Blaeser et al., 2024*).

## Results

### Characterizing macrophage $Ca^{2+}$ signaling features in homeostatic brain meninges

Previous intravital imaging studies exploring the spatiotemporal dynamics of tissue-resident macrophages have primarily used CX3C motif chemokine receptor 1 (CX3CR1)-based mouse reporter strains (*Rua et al., 2019*; *Mehari et al., 2022*; *Ren et al., 2023*; *Ishikawa-Ankerhold et al., 2024*). However, other resident monocyte-derived cells are labeled in these reporter mice (*Chinnery et al., 2010*; *Jung et al., 2000*). Moreover, brain microglia also express CX3CR1 (*Rua et al., 2019*; *Jung et al., 2000*; *Roth et al., 2014*; *Liu et al., 2019*), limiting the use of these reporter mice for resolving the spatiotemporal subcellular $Ca^{2+}$ dynamics of macrophages localized to the relatively thin meningeal layers covering the brain parenchyma. To systematically characterize meningeal macrophage $Ca^{2+}$ dynamics in vivo and avoid contamination from $Ca^{2+}$ signals arising from superficial parenchymal microglia, we leveraged recent findings showing that Pf4 is highly enriched in meningeal macrophages but not in other meningeal immunocytes or parenchymal microglia (*Van Hove et al., 2019*; *Pinho-Ribeiro et al., 2023*; *McKinsey et al., 2020*), and generated transgenic reporter mice expressing the highly sensitive

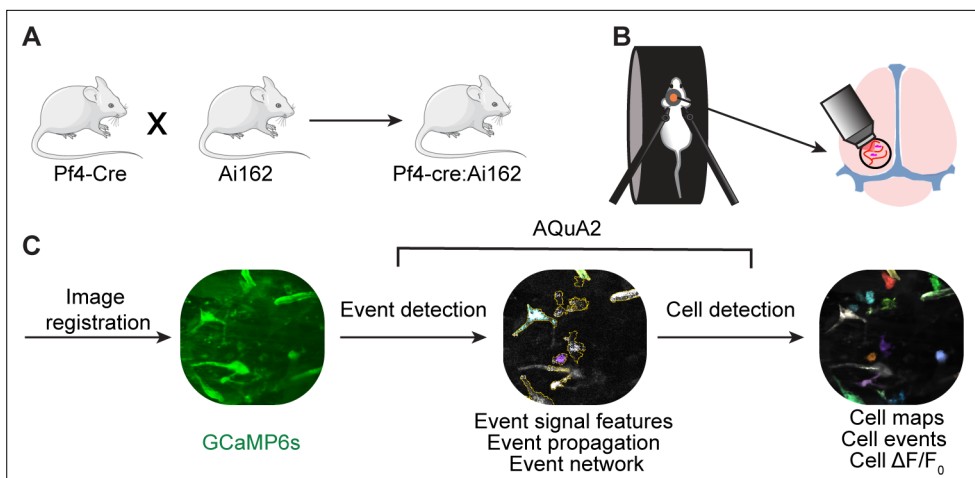

**Figure 1.** Imaging meningeal macrophage Ca²⁺ dynamics in awake behaving mice. (**A**) Pf4-Cre:Ai162, GCaMP6s reporter mouse construct for imaging meningeal macrophages Ca²⁺ activity. (**B**) Experimental procedure for two-photon imaging of meningeal macrophage Ca²⁺ activity. Following the implantation of a headpost and a cranial window, mice were habituated to head restraint and subjected to two-photon microscopy while head-fixed on a running wheel to study meningeal macrophage Ca²⁺ activity. (**C**) Macrophage Ca²⁺ imaging processing pipeline.

Ca²⁺ indicator GCaMP6s (using Ai162 mice; *Daigle et al., 2018*) in Pf4⁺ macrophages (using Pf4-Cre mice; *Tiedt et al., 2007*). Notably, 100% of meningeal macrophages are labeled in Pf4-Cre-based reporter mice (*McKinsey et al., 2020*).

Anesthetic agents impact intracellular Ca²⁺ signaling, including in brain macrophages and other non-excitable glial cells (*Umpierre et al., 2020*; *Thrane et al., 2012*; *Zhou et al., 2024*). We therefore investigated subcellular Ca²⁺ activity of meningeal macrophages in awake, behaving mice. We imaged meningeal macrophages via a chronic cranial window implanted together with a restraining headpost over the intact dura mater overlying the posterior neocortex. This chronic window approach produces minimal inflammatory responses in the cortex and meninges below the window (*Goldey et al., 2014*; *Blaeser et al., 2022*). After at least 7 days of recovery, mice were gradually habituated to head restraint over multiple days while free to run on a wheel (*Figure 1B*). We tracked Ca²⁺ transients of GCaMP6s-labeled meningeal macrophages using high-speed two-photon microscopy in 37 fields of view (FOVs) from 7 mice. We first corrected the imaging movies for locomotion-evoked meningeal translational shifts using rigid registration (*Blaeser et al., 2022*). Movies were then processed using the AQuA2 data analysis platform (*Mi et al., 2025*), which implements an unbiased event-based approach to capture spatiotemporal Ca²⁺ event dynamics (*Figure 1C*). Based on spatial analysis, we assigned events (*n* = 1361) with their corresponding Ca²⁺ features to each macrophage, using data from 503 cells.

Meningeal macrophages occupy two distinct niches: perivascular (along the abluminal surface, physically contacting mural cells) and non-perivascular (within the interstitial space) (*Amann et al., 2024*; *Min et al., 2024*). These spatial distributions may dictate their divergent roles in meningeal immunity and vascular regulation. We therefore characterized the Ca²⁺ dynamics of these two anatomically distinct macrophage populations (perivascular, *n* = 122; interstitial, non-perivascular, *n* = 381, respectively, *Figure 2A* and *Video 1*). Most perivascular macrophages (93.4%, *n* = 114) displaying ongoing Ca²⁺ activity were associated with vessels in the dura mater (labeled with a TRITC-Dextran tracer, see methods). The two meningeal macrophage subpopulations exhibited several distinct Ca²⁺ activity features. While the total area of Ca²⁺ activity in the peri- and non-perivascular macrophages was similar (*Figure 2B*), the signal perimeter of perivascular macrophages was significantly greater (*Figure 2C*) and exhibited a more elongated shape (*Figure 2A, D*), in agreement with their rod-shaped morphology (*Amann et al., 2024*; *Sato et al., 2021*). The Ca²⁺ event duration in the perivascular macrophages was also longer compared to the interstitial macrophage subpopulation (*Figure 2F*). The peak Ca²⁺ activity level (Max $\Delta F/F_0$, *Figure 2E*) and event rate (*Figure 2G*) were, nonetheless, similar in the two meningeal macrophage subpopulations.

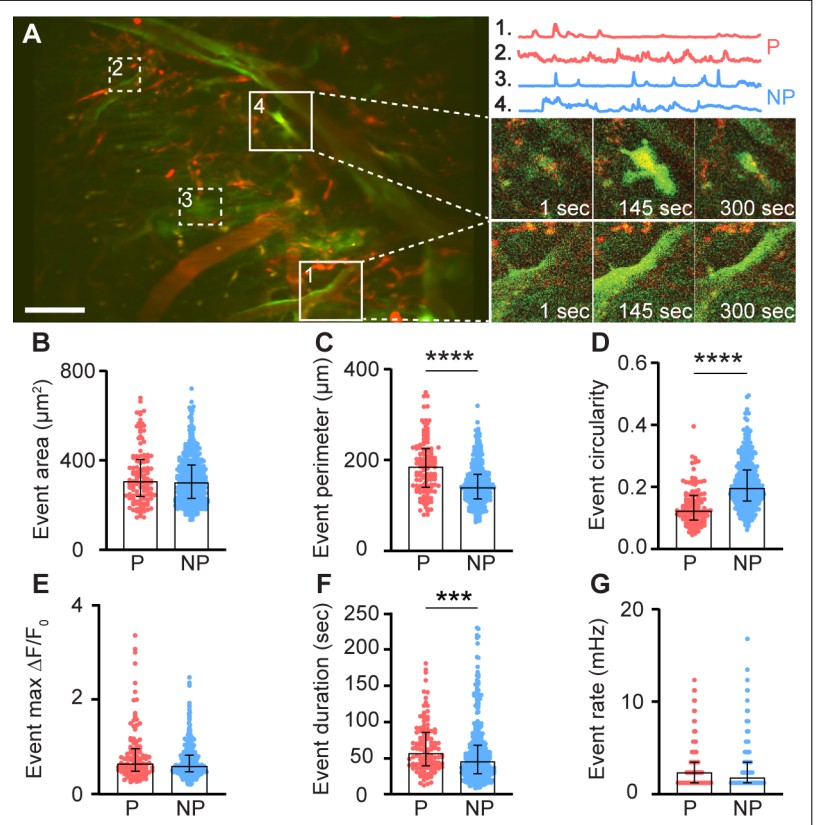

**Figure 2.** Ca$^{2+}$ dynamic features of meningeal macrophages at steady state. (**A**) Left: Mean projection of an example FOV depicting perivascular (P, red; 1, 2) and non-perivascular (NP, blue; 3, 4) meningeal macrophages (white squares). Scale bar: 50 μm. Right: Corresponding macrophages with representative 900-s Ca$^{2+}$ activity traces (top) and their fluorescence signal at selected time points (bottom). (**B–G**) AQuA2-based morphological and Ca$^{2+}$ event functional features of perivascular (p = 122 cells) and non-perivascular (NP, n = 381 cells) meningeal macrophage, n = 37 fields of view (FOVs) from 7 mice. (**B**) Event area, (**C**) event perimeter, (**D**) event circularity, (**E**) event max $\Delta F/F_0$, (**F**) event duration, and (**G**) event rate. Data (**B–G**) represent median ± IQR. ***p < 0.001, ****p < 0.0001, Mann–Whitney U-test.

---

Distinct Ca$^{2+}$ signal frequency spectra may underlie different biological functions of macrophages (*Mehari et al., 2022*). We therefore analyzed meningeal macrophage Ca$^{2+}$ signal waveforms using a multi-step signal processing and clustering analysis. We clustered cells based on two distinct patterns of Ca$^{2+}$ activity. Cells in Cluster 1 (n = 40) exhibited a more noisy-like activity pattern characterized by multiple frequencies, while cells in Cluster 2 (n = 463) displayed a single dominant frequency at 0.01 Hz (*Figure 3A*). To further explore these cellular signaling differences, we combined the clustering data with AQuA2-derived features and observed that macrophages in Cluster 1 showed a larger event perimeter, but lower circularity and peak magnitude than those in Cluster 2 (*Figure 3D–F*). While Cluster 1 cells had a lower signal-to-noise ratio compared to Cluster 2 cells (*Figure 3I*), both clusters displayed similar event area, duration, and rate (*Figure 3C, G, H*). Finally, we observed a significant association between cell cluster (1 vs. 2) and cell type, with Cluster 1 predominantly comprising perivascular

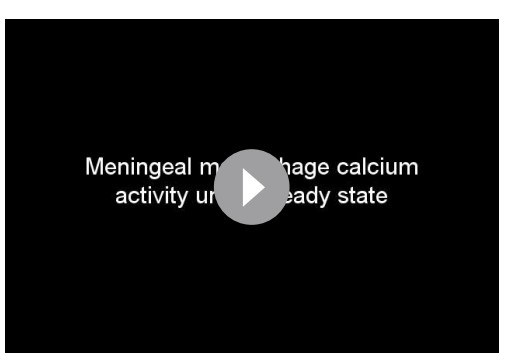

**Video 1.** Macrophage Ca$^{2+}$ dynamics in homeostatic meninges of awake mice. Example two-photon imaging showing Ca$^{2+}$ activity in perivascular and non-perivascular meningeal macrophages. Scale bar: 50 μm. https://elifesciences.org/articles/109888/figures#video1

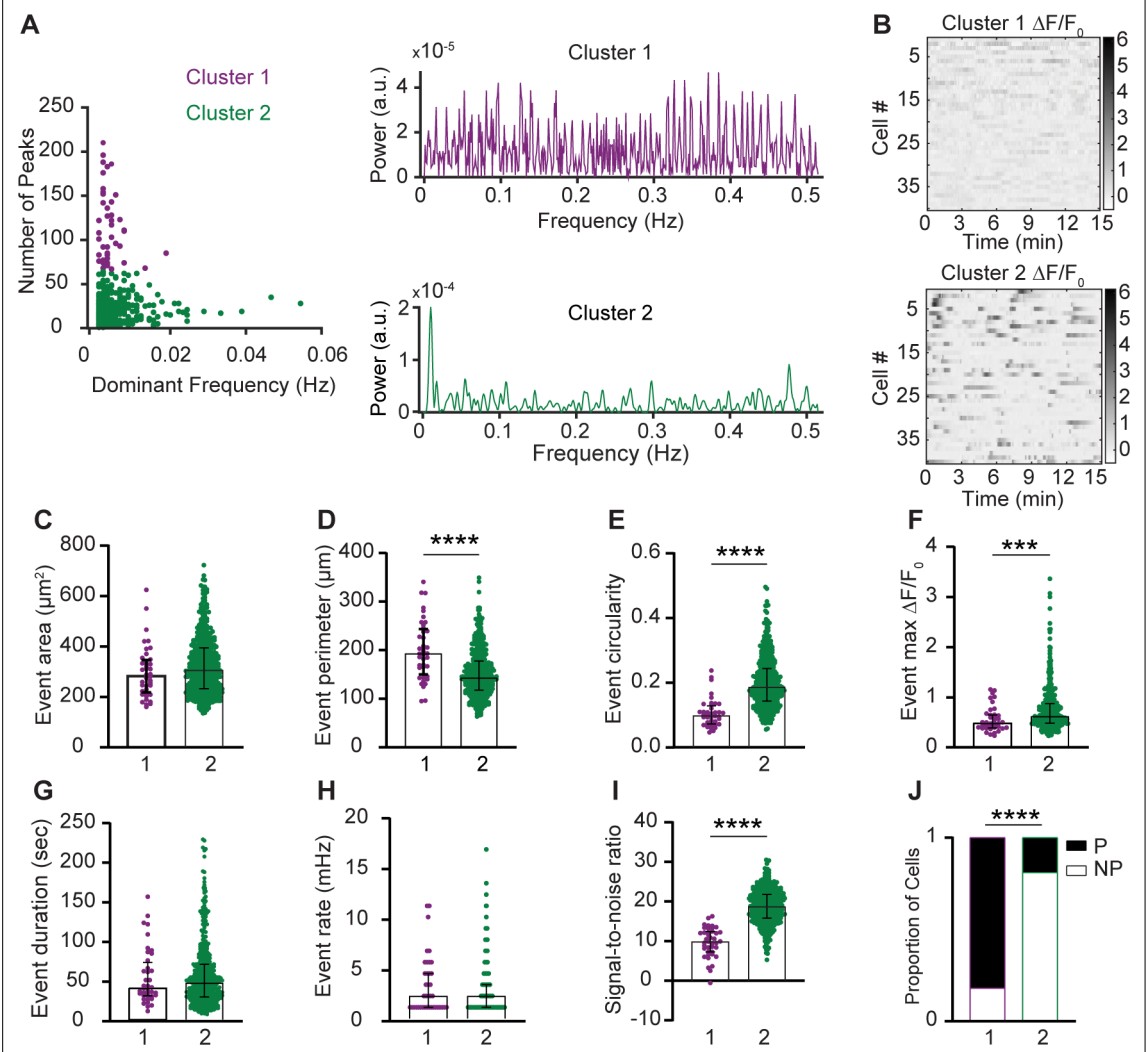

**Figure 3.** Intracellular Ca$^{2+}$ signal frequency spectra of macrophage subsets in the steady-state meninges. (**A**) Left: Clustering of meningeal macrophage Ca$^{2+}$ activity based on frequency-domain features and peak detection. Cluster 1 (purple, $n = 40$ cells) and Cluster 2 (green, $n = 463$ cells), $n = 37$ fields of view (FOVs) from 7 mice. Right: Power spectrum density (PSD) of Ca$^{2+}$ signals for each cluster. (**B**) Example $\Delta F/F_0$ heatmaps of Clusters 1 and 2 macrophages. (**C–H**) Morphological and Ca$^{2+}$ functional features of Clusters 1 and 2 macrophages. (**C**) Event area, (**D**) event perimeter, (**E**) event circularity, (**F**) event max $\Delta F/F_0$, (**G**) event duration, and (**H**) event rate. (**I**) Signal-to-noise ratio from Clusters 1 and 2. (**J**) Distribution of cell types across clusters. Data (**C–I**) represent median ± IQR. ***p < 0.001, ****p < 0.0001, Mann–Whitney U-test. Data (**J**) represent the cell proportion. ****p < 0.0001, Chi-square test.

macrophages and Cluster 2 comprising primarily non-perivascular macrophages (*Figure 3J*), further suggesting that these two meningeal macrophage subpopulations have distinct Ca$^{2+}$ signaling properties.

Intracellular Ca$^{2+}$ signal propagation underlies diverse cellular functions and has been recently identified in macrophages in vitro (*Taghdiri et al., 2021*) and in skin-resident macrophages in vivo (*Leon Guerrero et al., 2024*). By assessing the Ca$^{2+}$ signal propagation maps for each event within a defined cell, we identified two distinct patterns of activity: propagating events, in which Ca$^{2+}$ signals traveled throughout the entire cell, and stationary events (*Figure 4A*). Propagating events showed varied signal source regions and directionality. Most macrophages (perivascular, 94.3%, $n = 115$; non-perivascular, 86.9%, $n = 331$) exhibited only propagating events, while a small minority of cells displayed a mix of propagating and stationary events or exclusively stationary activity events (*Figure 4B*).

Macrophage intercellular communication, including synchronized activity, potentially across connected macrophage networks, has been implicated in maintaining tissue homeostasis and immune function (*Zumerle et al., 2019*; *Taghdiri et al., 2021*; *Paterson and Lämmermann, 2022*; *Behmoaras*

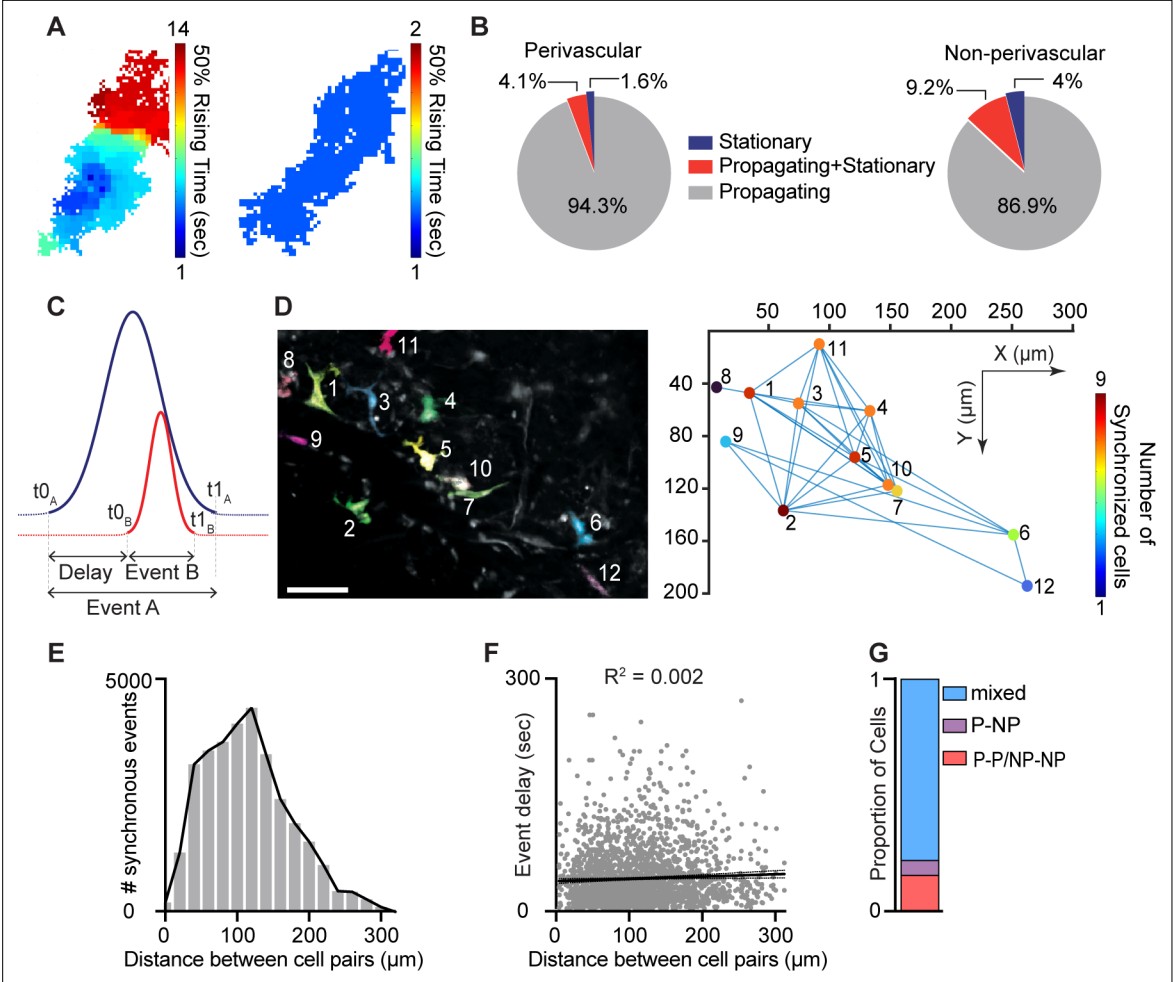

**Figure 4.** Intracellular propagation and intercellular synchronization of meningeal macrophage Ca²⁺ activity. (**A**) Spatial maps of two distinct Ca²⁺ events. Left: Propagating Ca²⁺ activity. Right: Stationary Ca²⁺ activity. (**B**) Distribution of event propagation profiles in perivascular (*n* = 122 cells) and non-perivascular (*n* = 381 cells) meningeal macrophages, *n* = 37 fields of view (FOVs) from 7 mice. (**C**) Schematic analysis paradigm for detecting synchronous Ca²⁺ activity in meningeal macrophages. (**D**) Synchronous Ca²⁺ events among meningeal macrophages within an FOV. Left: Mean projection of an example FOV showing Ca²⁺ activity in distinct macrophages (colored/numbered). Scale bar: 50 µm. Right: Spatial map of macrophages exhibiting synchronous Ca²⁺ activity. Lines connect macrophages with synchronized Ca²⁺ activity, and colors indicate the extent of Ca²⁺ event synchronization. (**E**) Distribution of distances across macrophage pairs showing different numbers of synchronous Ca²⁺ events. (**F**) Linear regression showing poor correlation between macrophage distances with synchronized Ca²⁺ activity and event delay. (**G**) Proportion of macrophages exhibiting a specific synchronized interaction (cells interacting only with the same subtype: P–P/NP–NP; cells interacting only with a different subtype: P–NP; mixed cells interacting with the same and different subtypes).

*et al., 2025*). To investigate meningeal macrophage intercellular interactions, we characterized spatio-temporal relationships between Ca²⁺ events in distinct cells within each FOV. We compared temporal factors, including the relative onset latency ($t0_A - t0_B$) between different macrophages and the duration of the first occurring event ($t1_A - t0_A$) (*Figure 4C*). We also compared the distances between macro-phage pairs exhibiting concurrent events and the number of synchronous events (*Figure 4D*). Finally, we calculated the proportion of perivascular and non-perivascular macrophages exhibiting synchro-nous Ca²⁺ events. Across all FOVs, 49.3% of macrophages exhibited co-activation over 0–300 µm with minimal distance–delay correlation (*Figure 4E, F*), suggesting that spatial proximity does not influence event synchronicity. Both macrophage subtypes exhibit temporally coincident Ca²⁺ eleva-tions (*Figure 4G*), consistent with a shared synchronization driver. The frequency of synchronous Ca²⁺ events detected could have been influenced by their duration (i.e., the longer the events, the higher the chance of detecting simultaneous event pairs). However, the duration of a given event was a poor predictor of the number of simultaneous events (*Figure 4H*).

## Dural perivascular macrophage Ca²⁺ activity is tuned to behaviorally driven dural vasomotion

Brain border-associated macrophages in the leptomeninges and related parenchymal perivascular spaces regulate pial arterial vasomotion indirectly by affecting vessel stiffness (*Drieu et al., 2022*). Yet, interaction between vascular-associated macrophages residing in the dura mater, the outermost meningeal layer, and dural vasomotion remains unknown. We therefore imaged the dural vasculature (using a TRITC-Dextran tracer) together with macrophage Ca²⁺ activity and used a generalized linear model (GLM) approach to investigate functional interaction between dural perivascular macrophage Ca²⁺ signals and dural vessel dynamics (*Figure 5A*). Dural arteries constrict during locomotion, while pial arteries dilate (*Gao and Drew, 2016*). We analyzed the locomotion-associated responses of 86 meningeal vessels (32 FOVs from 5 mice) and identified a subset (22%; $n = 19$) in which the diameter changes were well fit by a GLM with locomotion state as a predictor. Of these, we identified 74% ($n = 14$) as dural vessels based on their GLM's negative coefficients consistent with constriction (*Figure 5B, C*). Next, we fitted the Ca²⁺ signal observed in perivascular macrophages associated with these dural vessels ($n = 35$) to a GLM using the diameter changes as a predictor variable. Overall, the Ca²⁺ activity of 83% ($n = 29$) of these dural macrophages was well predicted by the model (average deviance explained across all well-fit macrophages: $0.43 \pm 0.15$, mean ± SD). Analysis of the macrophage–vascular models' beta coefficients revealed two distinct interactions. About half of the macrophages (55%, $n = 16$) exhibited negative coefficients (i.e., increase and decrease in Ca²⁺ activity associated with dural vasoconstriction and recovery, respectively; *Figure 5B, D*). The remaining macrophages (45%, $n = 13$) exhibited positive coefficients (i.e., decrease and increase in Ca²⁺ activity in response to dural constriction and recovery, respectively; *Figure 5B, E*). The coefficients for increased and decreased macrophage Ca²⁺ activity peaked near zero delay relative to the vasoconstriction and were not statistically different (*Figure 5D, E, H*). These data provide evidence that dural perivascular macrophages are functionally coupled to locomotion-driven dural vasomotion, either responding to or mediating it.

## An acute aberrant pro-inflammatory brain hyperexcitability event drives diverse Ca²⁺ dynamics in meningeal macrophages

CSD is a slowly propagating depolarization of neurons and astrocytes that drastically disrupts transmembrane gradients and cortical synaptic activity. This aberrant brain hyperexcitability event has been linked to parenchymal inflammation and pain in migraine, traumatic brain injury, and stroke (*Kaya et al., 2025*; *Harriott and Ayata, 2025*; *Levy and Moskowitz, 2023*), and could also affect meningeal macrophages (*Schain et al., 2018*). In anesthetized mice subjected to a single CSD episode, a small subset of meningeal macrophages undergoes morphological changes resembling an inflammatory state (*Kaya et al., 2025*). Given the direct anatomical and functional connections between the brain and meninges (*Smyth et al., 2024*; *Kipnis, 2024*) and the involvement of increased Ca²⁺ influx in macrophage inflammatory activation (*Chauhan et al., 2018*), we asked whether CSD drives intracellular Ca²⁺ elevations in meningeal macrophages. We used a pinprick stimulus in the frontal cortex to trigger a single CSD episode in awake mice (*Blaeser et al., 2024*; *Zhao and Levy, 2016*) and characterized the related changes in meningeal macrophage Ca²⁺ dynamics. In each experiment, we verified CSD induction based on the associated acute meningeal deformation and/or pial vasoconstriction observed in mice (*Blaeser et al., 2024* and *Video 2*). We studied CSD-related changes in Ca²⁺ dynamics in 249 macrophages (perivascular, $n = 64$; non-perivascular, $n = 185$; 13 FOVs from 10 mice). For each cell, we compared Ca²⁺ event rates during the passage of the CSD wave (1 min) and the PostCSD period (30 min) to baseline (PreCSD, 30 min) to assess acute and persistent Ca²⁺ responses, respectively. Given the low Ca²⁺ activity observed under steady state and the likelihood that no spontaneous Ca²⁺ elevations occur during the brief period of the CSD event, we considered cells to be either acutely activated or to exhibit an unchanged Ca²⁺ response (i.e., not activated). For studying more prolonged changes during the post-CSD period, we characterized cells as exhibiting persistently increased (event rate > 2× PreCSD), decreased (event rate < 0.5× PreCSD), or unchanged responses. These criteria were used to account for large, observable variations from baseline activity, while also minimizing the influence of spontaneous fluctuations observed in naïve mice. While consistent with previous studies on macrophages in different tissues (*Mehari et al., 2022*), these changes were not intended to represent definitive biological criteria for Ca²⁺ activation and inhibition, but rather a descriptive categorization based on comparable individual cell data.

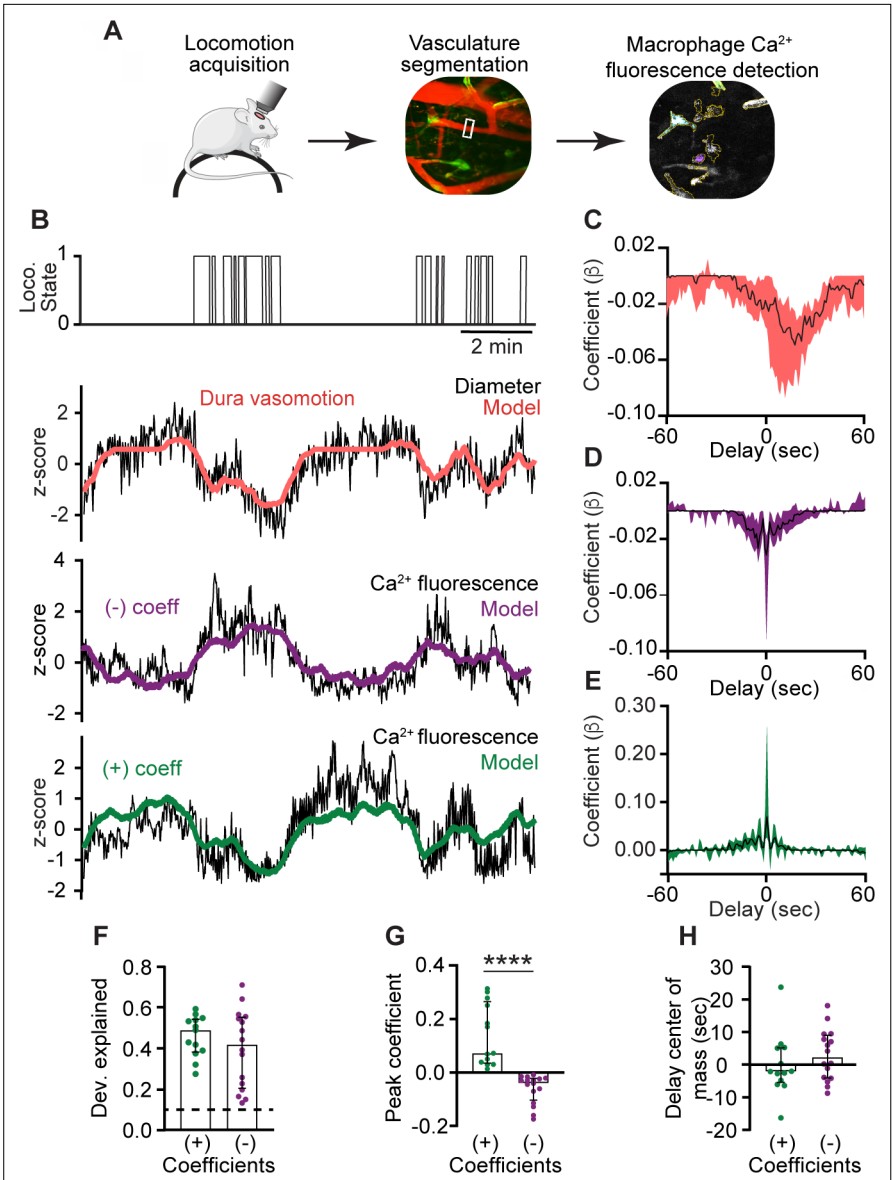

**Figure 5.** Ca²⁺ signals of dural perivascular macrophages are functionally coupled to behaviorally driven dural vasomotion. (**A**) Experimental paradigm: Locomotion data were acquired during imaging in awake-behaving mice. Behaviorally evoked changes in meningeal vessel diameter were obtained using segmentation of vessels labeled with a tracer during macrophage Ca²⁺ and further tested for coupling with Ca²⁺ signals. (**B**) Example data of macrophages with Ca²⁺ activity tuned to locomotion-related dural vessel vasomotion. Locomotion bouts (top trace) and dural vessel diameter (black trace) that were well fit by a generalized linear model (GLM; red line) using locomotion state as a predictor. Note vasoconstriction during locomotion, indicating a dural vessel. The two bottom traces depict the Ca²⁺ signals (black traces) of distinct meningeal macrophages, which were well fit by a GLM using dural vessel diameter as a predictor, with either a negative coefficient (purple) or a positive coefficient (green). (**C**) Temporal profile of dura vessels GLM coefficients averaged across all well-fitted vessels ($n$ = 19). Traces represent the median across all well-fit ROIs, with shaded regions indicating IQR. (**D, E**) Temporal profiles of GLM coefficient values for macrophages' Ca²⁺ activity averaged across all well-fitted cells (purple, negative coefficient, $n$ = 16; green, positive coefficient, $n$ = 13). Traces represent the median across all well-fit macrophages, with shaded regions indicating IQR. (**F**) The deviance explained (goodness-of-fit estimates) of the vessel diameter data included in the GLM used to predict macrophage Ca²⁺ activity were not statistically different for macrophages showing negative ($n$ = 16) and positive ($n$ = 13) coefficients, suggesting similar interaction levels. (**G**) Comparison of peak positive and negative coefficients of well-fit macrophage Ca²⁺ activity/vasomotion GLMs. (**H**) Center of mass of GLM coefficients indicating that dural vessel diameter changes drive bidirectional changes in fluorescence

*Figure 5 continued on next page*

*Figure 5 continued*

signal at zero delay. Data (**F–H**) are median ± IQR. ****p < 0.0001 (Mann–Whitney *U*-test). Data from 32 FOVs from 5 mice.

Using these criteria, we detected both acute and persistent $Ca^{2+}$ activity changes following CSD (*Figure 6A–E* and *Video 2*). While smaller subsets of meningeal macrophages exhibited acute (21.3%, *n* = 53) and/or persistent increases (22.1%, *n* = 55) in $Ca^{2+}$ activity; we observed a persistent decrease in the majority of cells (58.6%, *n* = 146). An acute increase was observed more often in peri-vascular macrophages (perivascular, 32.8%, *n* = 21; non-perivascular, 17.3%, *n* = 32, *Figure 6F*). Persistent changes in $Ca^{2+}$ activity were similarly observed in peri- and non-perivascular macrophages (increases; perivascular, 28.2%, *n* = 18; non-perivascular, 18.4%, *n* = 37; decreases; perivascular, 50.0%, *n* = 32; non-perivascular, 61.6%, *n* = 114, *Figure 6G*). The macrophages' propensity to develop a persistent $Ca^{2+}$ increase was unrelated to their acute response (*Figure 6H*), suggesting that the mechanisms underlying these two temporal responses are distinct. However, cells that showed no acute activation were more likely to exhibit decreased $Ca^{2+}$ activity post-CSD (*Figure 6H*). Finally, we observed that macrophages exhibiting a persistent increase in $Ca^{2+}$ activity had lower baseline activity than those showing a persistent decrease (*Figure 6I*), suggesting that this post-CSD response is influenced by the macrophages' basal $Ca^{2+}$ activity.

## CGRP receptor signaling mediates CSD-evoked persistent increase in meningeal macrophage $Ca^{2+}$ activity

Many meningeal macrophages are localized near peptidergic, CGRP-expressing sensory axons (*Pinho-Ribeiro et al., 2023*). In the wake of CSD, cortex-to-meninges signaling enhances the responsiveness of meningeal sensory neurons that could drive CGRP release from their peripheral nerve endings (*Blaeser et al., 2024*; *Zhao and Levy, 2016*; *Zhao and Levy, 2018a*). CGRP-expressing sensory neurons regulate tissue immunity and meningeal macrophage function via the CGRP neuroimmune axis (*Pinho-Ribeiro et al., 2023*; *Deng et al., 2024*). We therefore asked whether the CSD-related changes in meningeal macrophage $Ca^{2+}$ dynamics we observed involve CGRP receptor signaling. We pretreated mice with the selective CGRP receptor antagonist BIBN4096 and then imaged meningeal macrophage $Ca^{2+}$ activity (42 cells; perivascular, *n* = 14; non-perivascular, *n* = 28; 3 FOVs from 3 mice) before and after CSD. As expected, CGRP receptor antagonism did not affect CSD triggering (*Jin et al., 2025*). Compared with the control saline treatment, CGRP receptor blockade also did not reduce basal macrophage $Ca^{2+}$ activity (*Figure 7B*). Blocking CGRP receptor signaling neither affected the incidence of acute increases in $Ca^{2+}$ activity (*Figure 7A, C*) nor the magnitude of that response (*Figure 7D*). CGRP receptor antagonism, however, inhibited the CSD-evoked persistent increase in the macrophage's $Ca^{2+}$ activity, without affecting the incidence of the persistent decrease (*Figure 7E*). The data suggest that in the wake of CSD, the CGRP neuroimmune axis is responsible for the prolonged enhancement of $Ca^{2+}$ signaling in a subset of meningeal macrophages, which could potentially mediate their pro-inflammatory response.

## Discussion

Resident macrophages in the brain meninges are essential for maintaining brain homeostasis, regulating central nervous system immune surveillance, and mediating neuroimmune responses under pathological conditions (*Vara-Pérez and Movahedi, 2025*; *Rebejac et al., 2022*;

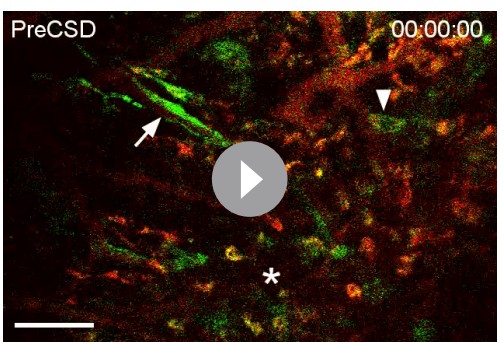

**Video 2.** Acute and persistent changes in meningeal macrophage $Ca^{2+}$ activity in response to cortical spreading depolarization (CSD). Example two-photon imaging of meningeal macrophage $Ca^{2+}$ activity at baseline, during, and following CSD. The arrow indicates a macrophage showing an acute $Ca^{2+}$ elevation, the arrowhead depicts a delayed and persistent $Ca^{2+}$ elevation, and the asterisk, a macrophage showing a persistent decreased $Ca^{2+}$ activity. Scale bar: 50 µm.
https://elifesciences.org/articles/109888/figures#video2

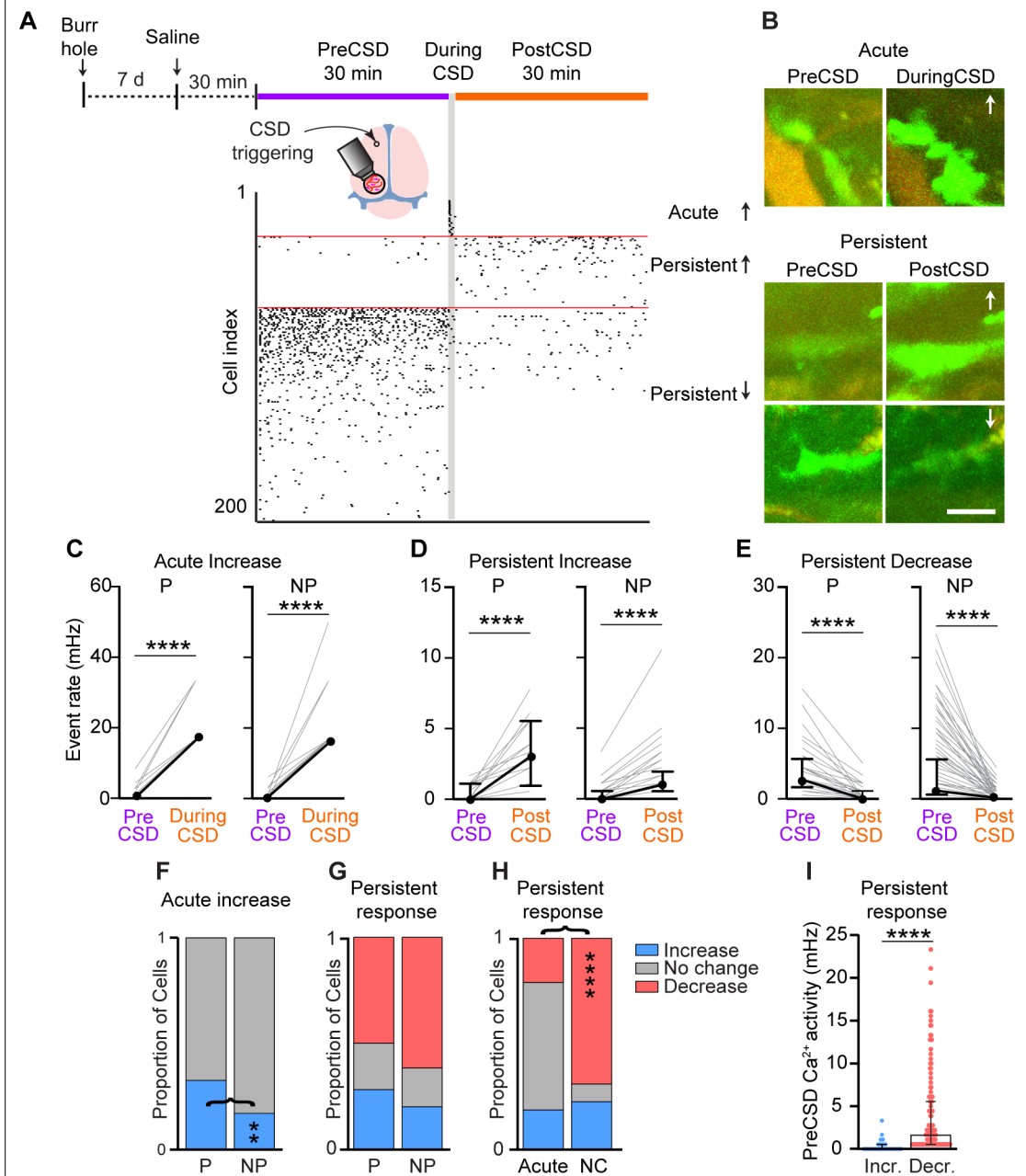

**Figure 6.** Diverse meningeal macrophage Ca²⁺ dynamics following cortical spreading depolarization (CSD). (**A**) Experimental setup and example data: (top) A small burr hole was drilled above the frontal cortex 7 days before Ca²⁺ imaging. Mice were pretreated with saline 30 min before imaging baseline macrophage Ca²⁺ activity (30 min, PreCSD). CSD was then induced with a pin prick, and macrophage Ca²⁺ activity was assessed during CSD (1 min during CSD) and post-CSD (30 min, post-CSD). (Bottom) A raster plot of macrophage Ca²⁺ activity showing the acute and persistent increases and persistent decrease in response to CSD. (**B**) Example of macrophage Ca²⁺ fluorescence changes following CSD. Images depict the mean projection over the specific experimental timeline. Arrows indicate an increase or a decrease in Ca²⁺ activity. Scale bar: 50 µm. (**C**) Individual responses of perivascular (P, n = 21) and non-perivascular (NP, n = 32) macrophages showing an acute increase in Ca²⁺ activity. (**D**) Individual response of P (n = 18) and NP (n = 37) macrophages exhibiting a persistent increase in Ca²⁺ activity. (**E**) Individual responses of P (n = 32) and NP (n = 114) macrophages showing a persistent decrease in Ca²⁺ activity. (**F**) Proportion of P and NP macrophages showing an acute increase in Ca²⁺ activity or no acute change. (**G**) Proportion of P and NP macrophages showing a persistent increase, decrease, or no change in Ca²⁺ activity. (**H**) Proportion of macrophages displaying distinct persistent responses stratified based on their acute response. (**I**) Baseline (PreCSD) Ca²⁺ activity in macrophages exhibiting persistent increased (n = 55) or decreased activity (n = 146). Data (**C–E**) are median ± IQR. ****p < 0.0001 (Wilcoxon signed rank test). Data (**F–H**) represent the proportion of cells; **p < 0.01; ****p < 0.0001 (Chi-square test). Data (**I**) are median ± IQR. ****p < 0.0001 (Mann–Whitney U-test). CSD data from n = 64 perivascular cells and n = 185 non-perivascular cells; 13 FOVs from 10 mice.

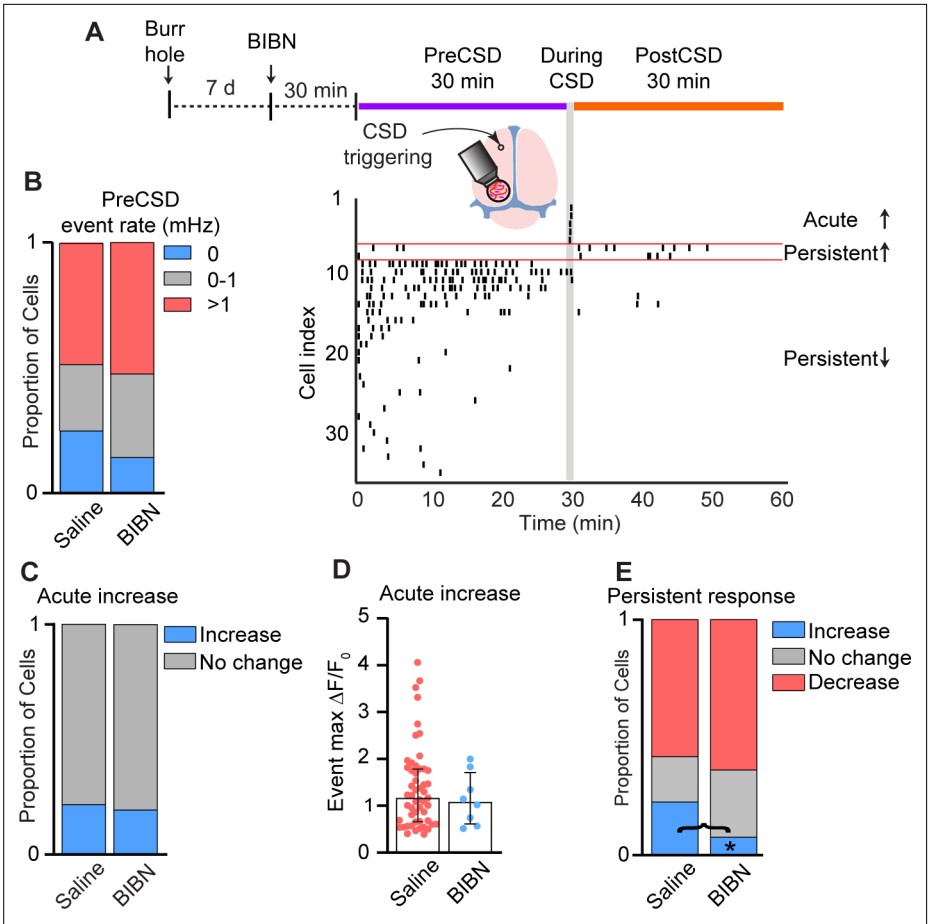

**Figure 7.** Calcitonin gene-related peptide (CGRP) receptor signaling mediates cortical spreading depolarization (CSD)-related persistent increase in meningeal macrophage Ca$^{2+}$ activity. (**A**) Experimental setup and example data: (top) A small burr hole was drilled above the frontal cortex 7 days before Ca$^{2+}$ imaging. Mice were pretreated with the CGRP receptor antagonist BIBN4096 (BIBN) 30 min before imaging baseline macrophage Ca$^{2+}$ activity (30 min, PreCSD). CSD was then induced with a pin prick, and macrophage Ca$^{2+}$ activity was assessed during CSD (1 min during CSD) and post-CSD (30 min, post-CSD). (Bottom) A raster plot of macrophage Ca$^{2+}$ activity showing the acute and persistent increases and persistent decrease in response to CSD. (**B**) CGRP receptor inhibition does not affect the baseline (PreCSD) event rate. Data compared between macrophages imaged in saline-treated mice ($n$ = 249 cells, 13 FOVs from 10 mice) and BIBN-treated mice ($n$ = 42 cells, 3 FOVs from 3 mice). (**C**) CGRP receptor antagonism does not affect the CSD-evoked acute increase in macrophage Ca$^{2+}$. Proportion of macrophages showing an acute response (increase vs. no change) in saline- and BIBN-treated mice. (**D**) CGRP receptor antagonism does not affect the magnitude of the acute macrophage Ca$^{2+}$ signal. Event max $\Delta F/F_0$ in macrophages showing an acute Ca$^{2+}$ increase in saline-treated mice ($n$ = 53 cells) and BIBN-treated mice ($n$ = 8 cells). (**E**) CGRP receptor antagonism distinctly inhibits the persistent increase in macrophage Ca$^{2+}$ activity post CSD. Proportion of macrophages showing persistent increase, persistent decrease, or no persistent change in saline-treated mice ($n$ = 249 cells) and BIBN-treated mice ($n$ = 42 cells). (**F**) Data (**B, C, E**) represent the proportion of cells. *$p$ < 0.05, Fisher's exact test. Data (**D**) are median ± IQR.

*Pinho-Ribeiro et al., 2023*; *Min et al., 2024*; *Rustenhoven and Kipnis, 2022*; *Dalmau Gasull et al., 2024*). Macrophages rely on Ca$^{2+}$ signaling to mediate many of their functions (*Desai and Leitinger, 2014*; *Zumerle et al., 2019*; *Seegren et al., 2020*; *Nascimento Da Conceicao et al., 2021*; *Taghdiri et al., 2021*; *Mehari et al., 2022*; *Seegren et al., 2023*), yet remarkably little is known about the Ca$^{2+}$ response properties of meningeal macrophages at steady state and disease. Here, using two-photon microscopy in awake behaving Pf4-Cre:Ai162, GCaMP6s reporter mice, we provide a foundational landscape of meningeal macrophage Ca$^{2+}$ dynamics. We describe a heterogeneity of meningeal macrophage Ca$^{2+}$ signals at steady state and in response to CSD, an aberrant cortical hyperexcitability event associated with migraine, traumatic brain injury, and stroke. Our data suggest that macrophages

in discrete perivascular and interstitial non-perivascular meningeal niches exhibit several distinct $Ca^{2+}$ signal properties at steady state. We further demonstrate $Ca^{2+}$ activity in dural perivascular macrophages, which is tuned to behaviorally driven dural vasomotion. Finally, we describe opposing $Ca^{2+}$ responses of meningeal macrophages following CSD, and demonstrate the contribution of CGRP receptor signaling in mediating CSD-evoked persistent $Ca^{2+}$ elevations.

The exact link between the distinct $Ca^{2+}$ signal properties of meningeal macrophage subsets observed herein and their homeostatic function remains to be established. The lower event magnitude and noisier signal observed in dural perivascular macrophages may reflect functional interactions with the pulsation dynamics of dural vessels (*Gao and Drew, 2016*). Indeed, by combining vascular and macrophage $Ca^{2+}$ imaging, we demonstrate a tight temporal association between the diameter fluctuations of the dural vessels and the $Ca^{2+}$ signal of their associated macrophages in awake, locomoting mice. Direct vascular–macrophage coupling may underlie this interaction, involving macrophages sensing vascular-related mechanical changes via Piezo1 signaling (*Atcha et al., 2021*). Functional vascular–macrophage interaction may also involve mural cells as intermediators (*Min et al., 2024*; *Tong et al., 2021*). Studying whether macrophage $Ca^{2+}$ signaling regulates dural vasomotion will require an experimental approach that has yet to be developed, enabling selective manipulation of perivascular dural macrophages. The paucity of ongoing $Ca^{2+}$ activity in perivascular macrophages situated in the leptomeninges we observed supports recent findings that subdural perivascular macrophages indirectly affect pial and parenchymal vasomotion via extracellular matrix remodeling (*Drieu et al., 2022*).

Intracellular $Ca^{2+}$ signal propagation has been observed in various non-excitable cells, such as astrocytes (*Semyanov and Verkhratsky, 2021*). We show that the majority of meningeal macrophages, including perivascular and interstitial cells, exhibit intracellular $Ca^{2+}$ signals that propagate throughout the entire cell, suggesting microdomain elevation of intracellular $Ca^{2+}$ following release from internal stores. By characterizing the spatiotemporal relationships between $Ca^{2+}$ signals in distinct cells, we also demonstrate synchronous events that are independent of spatial proximity, suggesting that synchronous $Ca^{2+}$ activity is not driven by intercellular communication. Further studies will be required to resolve the exact source of synchrony. Interestingly, our data indicate that synchronized events involve both peri- and non-perivascular macrophages, despite distinct $Ca^{2+}$ elevation patterns, suggesting that these meningeal macrophage subtypes similarly sense and respond to signals underlying synchronous activity.

Cortex-to-meninges signaling involves a relatively slow flow of soluble molecules within the cerebrospinal fluid that reach the subarachnoid space and then advance via arachnoid cuff exit points into the dura mater (*Smyth et al., 2024*; *Kipnis, 2024*). Our findings of acute $Ca^{2+}$ elevation in a subset of extrasinusoidal perivascular dural macrophages coinciding with the CSD event suggest a rapid transfer of soluble signaling factors released from a hyperexcitable cortex across all meningeal layers (*Levy and Moskowitz, 2023*). Nevertheless, we cannot exclude a mechanically driven macrophage response to the acute meningeal deformation produced by the neuronal and glial swelling and shrinkage of the cortical extracellular space during CSD (*Blaeser et al., 2024*; *Takano et al., 2007*; *Rosic et al., 2019*). Our data also indicate a delayed, prolonged increase in $Ca^{2+}$ signaling in a relatively small subset of macrophages post-CSD, which could underlie their pro-inflammatory-like morphological change (*Schain et al., 2018*; *Chauhan et al., 2018*). Our findings also support the view that meningeal neuroimmune CGRP signaling axis serves as a mechanism responsible for this macrophage $Ca^{2+}$ response, potentially via the activation of macrophage CGRP receptor complex (*Pinho-Ribeiro et al., 2023*) by CGRP released from sensitized meningeal afferent axons (*Blaeser et al., 2024*; *Zhao and Levy, 2016*; *Zhao and Levy, 2018b*). Whether the relatively small subset of meningeal macrophages featuring increased $Ca^{2+}$ signaling serves a protective role (*Lu et al., 2024*; *Hasegawa et al., 2024*) or a pro-inflammatory, destructive function (*Fattori et al., 2024*) remains to be elucidated. Intriguingly, our data point to a persistent decrease in macrophage $Ca^{2+}$ activity post-CSD, not involving CGRP receptor signaling, as the most prevalent response. Further studies are needed to determine whether this reduction in $Ca^{2+}$ activity reflects altered viability or reduced immune function that could interfere with the macrophage's ability to restore homeostasis and dampen local inflammation (*Rodríguez-Morales and Franklin, 2023*).

There are several limitations to our study. First, while PF4-Cre-based labeling has been shown to target brain border-associated macrophages, we cannot fully exclude the possibility that in a small

subset of meningeal dendritic cells, monocytes, and T cells that have low-level PF4 expression (*Van Hove et al., 2019*), GCaMP6 was also expressed, leading to a $Ca^{2+}$ signal. Nonetheless, a recent study using PF4-Cre:mTmG mice failed to detect EGFP reporter expression above background in any other meningeal cells by flow cytometry (*Barr et al., 2025*). Second, to enhance $Ca^{2+}$ event detection, we downsampled the movies to ~1 Hz. We therefore could have missed fast $Ca^{2+}$ transients or microdomain activity. Third, in our study, we imaged $Ca^{2+}$ activity in extrasinusoidal meningeal macrophages. It is therefore possible that these cells exhibit distinct response properties compared to the subset of dural macrophages associated with the dural sinuses (*Amann et al., 2024*). Finally, our study used a pharmacological approach to determine whether CGRP receptor signaling mediates macrophage $Ca^{2+}$ responses to CSD. We acknowledge that this approach does not allow us to establish a specific role for macrophage CGRP signaling, given the possibility that CGRP receptor signaling in other meningeal vascular or immune cells (*Van Hove et al., 2019*; *Pinho-Ribeiro et al., 2023*; *Monaghan et al., 2026*) may indirectly affect the macrophage $Ca^{2+}$ response.

## Conclusions

We provide a detailed characterization of macrophage $Ca^{2+}$ dynamics in homeostatic meninges, thereby expanding our understanding of their biological diversity. The coupling of dural perivascular macrophage $Ca^{2+}$ signals and dural vasomotion may represent a unique homeostatic functional dural macrophage–vascular unit that controls dural perfusion. The diversity of meningeal macrophage $Ca^{2+}$ responses to CSD further highlights the complexity of brain-to-meninges neuroimmune signaling and meningeal macrophage function in neurological disorders such as migraine, traumatic brain injury, and stroke. Our study also provides essential genetic and data analysis tools to further understand the molecular signaling underlying macrophage function at steady state and neuropathological conditions.

## Methods

### Animals

All experimental procedures were approved by the Beth Israel Deaconess Medical Center Institutional Animal Care and Use Committee (protocol # 072-2021-24). Experiments were conducted on adult Pf4-Cre:Ai162, GCaMP6s $Ca^{2+}$ reporter mice (8–17 weeks, 9 males, 5 females). Mice were generated by crossing Pf4-Cre mice [C57BL/6-Tg (Pf4-icre) Q3Rsko/J, Jackson laboratory, Strain #008535] with Ai162 mice (B6.Cg-Igs7$^{tm162.1(tetO-GCaMP6s,CAG-tTA2)Hze}$/J, Jackson laboratory, Strain #031562). Animals were genotyped by Transnetyx Inc.

### Surgical procedures

Animals were anesthetized using isoflurane in 100% $O_2$ (induction: 3%; maintenance: 1.5–2%) and placed on a heating pad with a rectal probe attached to a stereotaxic frame to monitor animal body temperature during surgery. Animals received dexamethasone (8 mg/kg, i.p.) and Meloxicam SR (4 mg/kg, s.c.) to reduce inflammation and improve surgical outcomes. An eye ointment was used to prevent ocular drying. Mice were implanted with a titanium headpost and a 3-mm glass cranial window (1.5 mm lateral and 2 mm posterior to Bregma) over an intact dura covering the left posterior neocortex (*Blaeser et al., 2024*). Immediately after surgery, the mouse cage was placed on a water-circulating heating pad for faster recovery. Animals were then single-housed with access to a running wheel and a hut and allowed to recover for at least 1 week.

### Wheel running acclimation

After the cranial window surgery, mice were allowed to recover for at least a week. To reduce stress associated with head-fixation during imaging and habituate to wheel running, the mice received multiple training sessions (10 min to 1 hr over 3–4 days). In each session, the mouse was placed on a 3D printed running wheel, with its headpost attached to two clamps, and allowed to locomote freely.

### Two-photon $Ca^{2+}$ imaging

Awake-behaving mice were head-fixed to the running wheel by its headpost (*Figure 1*). We used a two-photon microscope (Neurolabware) with a Nikon 16X, 0.8 N.A. objective to acquire images at 15.5 Hz with 4X digital zoom (312 × 212 μm² FOV). A MaiTai laser set to 920 nm with 25–40 mW

power was used to excite fluorescence. The Scanbox package for MATLAB (Neurolabware) was used to control the microscope and acquire images and wheel running data. To image the meningeal vasculature, mice were administered 70 kDa TRITC-Dextran tracer (50 mg/kg, i.v.; Sigma-Aldrich).

### Behavioral tracking during two-photon imaging

We recorded running speed in MATLAB using a custom-made encoder (Arduino) coupled to the 3D-printed running wheel. See below for details of analyses of behavioral variables.

### Induction of CSD and pharmacological treatment

For CSD induction, a 1-mm burr hole was drilled at the frontal bone (1.5 mm anterior to the cranial window) to allow access to the brain cortical surface. A small amount of silicone elastomer (Kwik-Cast, WPI) was placed to cover the burr hole opening, and the animal was left to recover for at least 1 week. CSD was induced using a brief 2-s cortical pinprick (*Blaeser et al., 2024*). CSD induction was confirmed by the identification of a short-lasting meningeal deformation and/or transient pial constriction (*Blaeser et al., 2024*). On the experimental day, 30 min before baseline recording, mice were pretreated with the selective CGRP receptor antagonist BIBN4096 (0.3 ml, 1 mg/kg, i.p., Tocris) (*Zhao and Levy, 2018a*) or 0.3 ml of saline (Vehicle control).

### Quantification and statistical analysis

#### Two-photon imaging movie processing

We used a discrete Fourier transform to perform rigid registration to correct for translation changes caused by brain motion during locomotion. Movies were then downsampled to 1.03 Hz. Locomotion signals were detected as described (*Blaeser et al., 2022*). All image processing and locomotion signal extraction were performed in MATLAB 2021b (Mathworks).

#### $Ca^{2+}$ signal detection pipeline

For detecting macrophage $Ca^{2+}$ signals, we used the Activity Quantification and Analysis (AQuA2) platform that implements an event-based approach with advanced machine learning techniques for temporal and spatial segmentation of $Ca^{2+}$ fluorescence events (*Tiedt et al., 2007*). Importantly, this computational platform captures event dynamics beyond traditional ROI-based approaches. The following user-defined parameters were input: 0.49 μm/pixel spatial resolution, 1.03 Hz temporal resolution, 1-s minimal event duration detection; window of event size between 25% (125 μm²/ 523 pixels) and 150% (750 μm²/ 3,140 pixels). Every event and cell identified was followed by a manual visual check. Subsequently, each cell was labeled as perivascular or non-perivascular according to its location relative to vessels. Morphological features (area, perimeter, circularity) and spatiotemporal aspects of the $Ca^{2+}$ signals (i.e., $\Delta F/F_0$ dynamics, frequency, amplitude, duration) were used to analyze cell-specific characteristics and $Ca^{2+}$ activity profiles. We employed the AQuA2 automatically generated function (*Mi et al., 2025*) to characterize intracellular $Ca^{2+}$ propagation. Intercellular $Ca^{2+}$ activity was evaluated by analyzing temporally co-occurring events (synchronized event pairs), as well as their corresponding spatial localization in the FOV. All post-processing of AQuA2-generated data was performed using MATLAB 2021b (Mathworks).

#### Clustering of $Ca^{2+}$ dynamics

We used a Savitzky–Golay filter to detrend and smooth $Ca^{2+}$ activity traces. Polynomial order and frame size were optimized for each cell by selecting the parameter combination that yielded the best signal-to-noise ratio. From the filtered signals, we extracted the dominant frequency using Fast Fourier Transform, and peak counts using minimum peak prominence (threshold of 10% signal amplitude range). The optimal number of clusters was determined using the Elbow method, after which k-means clustering was applied to group cells based on their signal characteristics.

#### Analysis of CSD-related changes in macrophage $Ca^{2+}$ dynamics

For analyzing the effects of CSD on macrophages' $Ca^{2+}$ dynamics, we divided the activity of each cell into three phases: 'PreCSD' (0–30 min), 'DuringCSD' (30–31 min), and 'PostCSD' (31–61 min). $Ca^{2+}$ event rates during CSD and PostCSD were compared with those of the PreCSD baseline to evaluate

acute and persistent Ca$^{2+}$ responses, respectively. Ca$^{2+}$ responses were categorized as increased (event rate >2x PreCSD), decreased (event rate <0.5x PreCSD), or unchanged.

### Analysis of locomotion

We extracted running speed (cm/s) from the wheel encoder. To infer the locomotion state, we first concatenated all velocity signals obtained from a given mouse across all experiments and trained a two-state Hidden Markov Model using the MATLAB function 'hmmtrain'. Then, the locomotion state was inferred for each individual imaging run by applying the MATLAB function 'hmmviterbi' with the model trained on the concatenated data. Locomotion bouts were defined as periods when the locomotion state was sustained for at least 2 s.

### Vascular signals

We calculated changes in vascular diameter by first generating a maximum-intensity projection of the red channel and then drawing polygons (ROIs) around each vessel. Subsequently, for each frame, pixels inside each ROI were extracted, and a Radon transform was applied to get a 1D vessel profile. Radon transform of the first frame was used as a reference to normalize the results of subsequent frames, resulting in a time series of normalized diameter traces.

### General linear models

We investigate the functional interaction between dural perivascular macrophage Ca$^{2+}$ activity and dural vasomotion by fitting a Gaussian GLM using the MATALB GLMnet package MATLAB 2021b, with elastic net regularization ($\alpha$ = 0.01) and 10-fold cross-validation. We employed a two-step modeling. The first model was used to identify dural vessels by evaluating the correlation between changes in vessel diameter and locomotion state, and classifying dural or pial vessels based on their vasoconstriction and vasodilation dynamics, respectively (*Gao and Drew, 2016*). The second modeling step evaluated the relative contribution of perivascular dural Ca$^{2+}$ activity signals to changes in dural vessel diameter. All signals were downsampled to 1.03 Hz to match the frame rate used when extracting fluorescence Ca$^{2+}$ signals in AQuA2. To allow potential anticipatory or delayed responses of diameter-to-locomotion or Ca$^{2+}$ fluorescence-to-diameter, the time interval used to analyze the response to the predictor was set from –60 to +60 s. The GLM was trained on 75% of the data, and all predictions and model performance reported are from the remaining 25% testing set. A threshold of 0.1 goodness-of-fit deviance explained was set. For each predictor temporal shift, a response coefficient was generated. The centroid delay between the predictor and response was calculated, weighted by the absolute value of the coefficient (i.e., center-of-mass).

### Statistical analysis

All statistical analyses were performed using GraphPad Prism 10.4 and MATLAB 2021b. Data were analyzed using a Wilcoxon matched-pairs signed rank sum test or a Mann–Whitney *U*-test. Distribution of categorical data was analyzed using the Chi-square or Fisher's exact test. p-values are indicated as follows: *$p < 0.05$, **$p < 0.01$, ***$p < 0.001$, ****$p < 0.0001$.

## Acknowledgements

The study was supported by NIH grants: R21NS130561; R01NS115972 and R01NS133625 to DL.

## Additional information

### Funding

| Funder | Grant reference number | Author |
| --- | --- | --- |
| National Institutes of Health | R21NS130561 | Dan Levy |

| Funder | Grant reference number | Author |
| --- | --- | --- |
| National Institutes of Health | R01NS115972 | Dan Levy |
| National Institutes of Health | R01NS133625 | Dan Levy |

The funders had no role in study design, data collection, and interpretation, or the decision to submit the work for publication.

## Author contributions

Simone Carneiro-Nascimento, Data curation, Software, Formal analysis, Investigation, Writing – original draft, Writing – review and editing; Chao Wei, Anna Gutterman, Data curation; Dan Levy, Conceptualization, Formal analysis, Supervision, Funding acquisition, Investigation, Writing – original draft, Project administration, Writing – review and editing

## Author ORCIDs

Simone Carneiro-Nascimento https://orcid.org/0000-0002-6141-6080
Dan Levy https://orcid.org/0000-0003-0630-6660

## Ethics

This study was performed in strict accordance with the recommendations in the Guide for the Care and Use of Laboratory Animals of the National Institutes of Health. All experimental procedures were approved by the Beth Israel Deaconess Medical Center Institutional Animal Care and Use Committee (protocol # 072-2021-24).

Reviewer #1 (Public review): https://doi.org/10.7554/eLife.109888.3.sa1
Reviewer #2 (Public review): https://doi.org/10.7554/eLife.109888.3.sa2
Reviewer #3 (Public review): https://doi.org/10.7554/eLife.109888.3.sa3
Author response https://doi.org/10.7554/eLife.109888.3.sa4

# Additional files

## Supplementary files
MDAR checklist

## Data availability

All data needed to evaluate the conclusions in the paper are present in the manuscript. The code used for analyzing the data in this study was deposited in the Levy Lab GitHub account. Code for movie processing is available at GitHub (copy archived at *Levy Lab_Headache, 2026a*). Code for locomotion analysis is available GitHub (copy archived at *Levy Lab_Headache, 2026b*). Code for post Aqua2-processing is available at GitHub (copy archived at *Levy Lab_Headache, 2026c*). Code for vascular segmentation is available at GitHub (copy archived at *Levy Lab_Headache, 2026d*). Code for GLM is available at GitHub (copy archived at *Levy Lab_Headache, 2026e*).

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
