## [Editor Report · eLife Assessment]

This study provides **important** insights into how immune cells in the brain's protective layers behave under normal and disease-like conditions, revealing location-specific activity patterns that may shape inflammation and disorders such as migraine. The evidence is **compelling** and supported by advanced imaging approaches and rigorous analyses, although some conceptual and interpretational limitations temper the mechanistic depth. Overall, the work will be of broad interest and represents an invaluable contribution to the growing field linking immune and nervous system function.

---

## [Referee Report · Reviewer #1 (Public review)]

Summary:

This study presents a technically sophisticated intravital two-photon calcium imaging approach to characterize meningeal macrophage Ca^2+^ dynamics in awake mice. The development of a Pf4Cre:GCaMP6s reporter line and the integration of event-based Ca^2+^ analysis represent clear methodological strengths. The findings reveal niche-specific Ca^2+^ signaling patterns and heterogeneous macrophage responses to cortical spreading depolarization (CSD), with potential relevance to migraine and neuroinflammatory conditions. Despite these strengths, several conceptual, technical, and interpretational issues limit the impact and mechanistic depth of the study. Addressing the points below would substantially strengthen the manuscript.

Strengths:

The use of chronic two-photon Ca^2+^ imaging in awake, behaving mice represents a major technical strength, minimizing confounds introduced by anesthesia. The development of a Pf4Cre:GCaMP6s reporter line, combined with high-resolution intravital imaging, enables long-term and subcellular analysis of macrophage Ca^2+^ dynamics in the meninges.

The comparison between perivascular and non-perivascular macrophages reveals clear niche-dependent differences in Ca^2+^ signaling properties. The identification of macrophage Ca^2+^ activity temporally coupled to dural vasomotion is particularly intriguing and highlights a potential macrophage-vascular functional unit in the dura.

By linking macrophage Ca^2+^ responses to CSD and implicating CGRP/RAMP1 signaling in a subset of these responses, the study connects meningeal macrophage activity to clinically relevant neuroimmune pathways involved in migraine and other neurological disorders.

Weaknesses:

The manuscript relies heavily on Pf4Cre-driven GCaMP6s expression to selectively image meningeal macrophages. Although prior studies are cited to support Pf4 specificity, Pf4 is not an exclusively macrophage-restricted marker, and developmental recombination cannot be excluded. The authors should provide direct validation of reporter specificity in the adult meninges (e.g., co-labeling with established macrophage markers and exclusion of other Pf4-expressing lineages). At minimum, the limitations of Pf4Cre-based labeling should be discussed more explicitly, particularly regarding how off-target expression might affect Ca^2+^ signal interpretation.

The manuscript offers an extensive characterization of Ca^2+^ event features (frequency spectra, propagation patterns, synchrony), but the biological significance of these signals is largely speculative. There is no direct link established between Ca^2+^ activity patterns and macrophage function (e.g., activation state, motility, cytokine release, or interaction with other meningeal components). The discussion frequently implies functional specialization based on Ca^2+^ dynamics without experimental validation. To strengthen the conceptual impact, a clearer framing of the study as a foundational descriptive resource, rather than a functional dissection, would improve alignment between data and conclusions.

The GLM analysis revealing coupling between dural perivascular macrophage Ca^2+^ activity and vasomotion is technically sophisticated and intriguing. However, the directionality of this relationship remains unresolved. The current data do not distinguish whether macrophages actively regulate vasomotion, respond to mechanical or hemodynamic changes, or are co-modulated by neural activity. Statements suggesting that macrophages may "mediate" vasomotion are therefore premature. The authors should reframe these conclusions more cautiously, emphasizing correlation rather than causation, and expand the discussion to explicitly outline experimental strategies required to establish causality (e.g., macrophage-specific Ca^2+^ manipulation).

The authors conclude that synchronous Ca^2+^ events across macrophages are driven by extrinsic signals rather than intercellular communication, based primarily on distance-time analyses. This conclusion is not sufficiently supported, as spatial independence alone does not exclude paracrine signaling, vascular cues, or network-level coordination. No perturbation experiments are presented to test alternative mechanisms. The authors can either provide additional experimental evidence or rephrase the conclusion to acknowledge that the source of synchrony remains unresolved.

A major and potentially important finding is that the dominant macrophage response to CSD is a persistent decrease in Ca^2+^ activity, which is independent of CGRP/RAMP1 signaling. However, this phenomenon is not mechanistically explored. It remains unclear whether Ca^2+^ suppression reflects macrophage inhibition, altered viability, homeostatic resetting, or an anti-inflammatory program. Minimally, the discussion should be more deeply engaged with possible interpretations and implications of this finding.

The pharmacological blockade of RAMP1 supports a role for CGRP signaling in persistent Ca^2+^ increases after CSD, but the experiments are based on a relatively small number of cells and animals. The limited sample size constrains confidence in the generality of the conclusions. Pharmacological inhibition alone does not establish cell-autonomous effects in macrophages. The authors should acknowledge these limitations more explicitly and avoid overextension of the conclusions.

Comments on revisions:

The authors have answered the questions well.

---

## [Referee Report · Reviewer #2 (Public review)]

Using chronic intravital two-photon imaging of calcium dynamics in meningeal macrophages in Pf4Cre:TIGRE2.0-GCaMP6 mice, the study identified heterogeneous features of perivascular and non-perivascular meningeal macrophages at steady state and in response to cortical spreading depolarization (CSD). Analyses of calcium dynamics and blood vessels revealed a subpopulation of perivascular meningeal macrophages whose activity is coupled to behaviorally driven diameter fluctuations of their associated vessels. The analyses also investigated synchrony between different macrophage populations and revealed a role for CGRP/RAMP1 signaling in the CSD-induced increase, but not the decrease, in calcium transients.

This is a timely study at both the technical and conceptual levels, examining calcium dynamics of meningeal macrophages in vivo. The conclusions are well supported by the findings and will provide an important foundation for future research on immune cell dynamics within meninges in vivo. The paper is well written and clearly presented.

---

## [Referee Report · Reviewer #3 (Public review)]

Summary:

The authors of this report wish to show that distinct populations of meningeal macrophages respond to cortical spreading depolarization (CSD) via unique calcium activity patterns depending on their location in the meningeal sub compartments. Perivascular macrophages display calcium signaling properties that are sometimes in opposition to non-perivascular macrophages. Many of the meningeal macrophages also displayed synchronous activity at variable distances from one another. Other macrophages were found to display calcium signals in response to dural vasomotion. CSD could induce variable calcium responses in both perivascular and non-perivascular macrophages in the meninges in part due to RAMP1 dependent effects. Results will inform future research on the calcium responses displayed by macrophages in the meninges under both normal and pathological conditions.

Strengths:

Sophisticated in vivo imaging of meningeal immune cells is employed in the study which has not been performed previously. A detailed analysis of the distinct calcium dynamics in various subtypes of meningeal macrophages is provided. Functional relevance of the responses are also noted in relation to CSD events.

Weaknesses:

Specificity of the methods used to target both meningeal macrophages and RAMP1 are limited. A discussion section on potential pitfalls is included to address this.

---

## [Author Response]

The following is the authors’ response to the original reviews.

**Public review:**

**Reviewer #1 (Public review):**
Strengths:(1) The use of chronic two-photon Ca^2+^ imaging in awake, behaving mice represents a major technical strength, minimizing confounds introduced by anesthesia. The development of a Pf4Cre:GCaMP6s reporter line, combined with high-resolution intravital imaging, enables long-term and subcellular analysis of macrophage Ca^2+^ dynamics in the meninges.(2) The comparison between perivascular and non-perivascular macrophages reveals clear niche-dependent differences in Ca^2+^ signaling properties. The identification of macrophage Ca^2+^ activity temporally coupled to dural vasomotion is particularly intriguing and highlights a potential macrophage-vascular functional unit in the dura.3) By linking macrophage Ca^2+^ responses to CSD and implicating CGRP/RAMP1 signaling in a subset of these responses, the study connects meningeal macrophage activity to clinically relevant neuroimmune pathways involved in migraine and other neurological disorders.

Thank you for recognizing the strengths in our work.

Weaknesses:(1) The manuscript relies heavily on Pf4Cre-driven GCaMP6s expression to selectively image meningeal macrophages. Although prior studies are cited to support Pf4 specificity, Pf4 is not an exclusively macrophage-restricted marker, and developmental recombination cannot be excluded. The authors should provide direct validation of reporter specificity in the adult meninges (e.g., co-labeling with established macrophage markers and exclusion of other Pf4-expressing lineages). At minimum, the limitations of Pf4Cre-based labeling should be discussed more explicitly, particularly regarding how off-target expression might affect Ca^2+^ signal interpretation.

We acknowledge that PF4 is not an exclusively macrophage-restricted marker. Yet, among meningeal immunocytes, it is almost exclusively expressed in macrophages (1, 2). Furthermore, in the adult mouse meninges, PF4^Cre^-based reporter lines label nearly all dural and leptomeningeal macrophages and almost no other cells (3, 4). This Cre line has also been used to target border-associated macrophages (2, 4). Moreover, a recent study suggests that the bacterial artificial chromosome used to generate the PF4^Cre^ line does not affect meningeal macrophage activity (4). Nonetheless, in the revised version, we discuss a potential limitation of the Pf4Cre-based labeling approach for studying meningeal macrophages’ Ca^2+^ signaling, namely that a very small population of other meningeal immune cells may also be labeled.

(2) The manuscript offers an extensive characterization of Ca^2+^ event features (frequency spectra, propagation patterns, synchrony), but the biological significance of these signals is largely speculative. There is no direct link established between Ca^2+^ activity patterns and macrophage function (e.g., activation state, motility, cytokine release, or interaction with other meningeal components). The discussion frequently implies functional specialization based on Ca^2+^ dynamics without experimental validation. To strengthen the conceptual impact, a clearer framing of the study as a foundational descriptive resource, rather than a functional dissection, would improve alignment between data and conclusions.

In our discussion, we indicated that “the exact link between the distinct Ca^2+^ signal properties of meningeal macrophage subsets observed herein and their homeostatic function remains to be established”. In the revised discussion part, we acknowledge that this is primarily a descriptive study that provides a foundational landscape of Ca^2+^ dynamics in meningeal macrophages.

(3) The GLM analysis revealing coupling between dural perivascular macrophage Ca^2+^ activity and vasomotion is technically sophisticated and intriguing. However, the directionality of this relationship remains unresolved. The current data do not distinguish whether macrophages actively regulate vasomotion, respond to mechanical or hemodynamic changes, or are co-modulated by neural activity. Statements suggesting that macrophages may "mediate" vasomotion are therefore premature. The authors should reframe these conclusions more cautiously, emphasizing correlation rather than causation, and expand the discussion to explicitly outline experimental strategies required to establish causality (e.g., macrophage-specific Ca^2+^ manipulation).

In the results section, we indicate that our data suggest that dural perivascular macrophages are functionally coupled to locomotion-driven dural vasomotion, either responding to it or mediating it. Furthermore, we discussed the possibilities that (1) macrophages sense vascular-related mechanical changes and (2) macrophage Ca^2+^ signaling regulates dural vasomotion. Moreover, we explicitly state that studying causality will require an experimental approach that has yet to be developed, enabling selective manipulation of dural perivascular macrophages.

(4) The authors conclude that synchronous Ca^2+^ events across macrophages are driven by extrinsic signals rather than intercellular communication, based primarily on distance-time analyses. This conclusion is not sufficiently supported, as spatial independence alone does not exclude paracrine signaling, vascular cues, or network-level coordination. No perturbation experiments are presented to test alternative mechanisms. The authors can either provide additional experimental evidence or rephrase the conclusion to acknowledge that the source of synchrony remains unresolved.

Thank you for this suggestion. In the revision, we indicate that further studies are required to resolve the exact source of synchrony.

(5) A major and potentially important finding is that the dominant macrophage response to CSD is a persistent decrease in Ca^2+^ activity, which is independent of CGRP/RAMP1 signaling. However, this phenomenon is not mechanistically explored. It remains unclear whether Ca^2+^ suppression reflects macrophage inhibition, altered viability, homeostatic resetting, or an anti-inflammatory program. Minimally, the discussion should be more deeply engaged with possible interpretations and implications of this finding.

While we propose that the decrease in macrophage Ca^2+^ signaling following CSD could indicate that a hyperexcitable cortex dampens meningeal immunity, in the revised discussion, we indicate that further studies are needed to determine whether this reduction in meningeal macrophage Ca^2+^ activity reflects altered viability or reduced immune function that could interfere with the macrophage’s ability to restore homeostasis and dampen local inflammation.

(6) The pharmacological blockade of RAMP1 supports a role for CGRP signaling in persistent Ca^2+^ increases after CSD, but the experiments are based on a relatively small number of cells and animals. The limited sample size constrains confidence in the generality of the conclusions. Pharmacological inhibition alone does not establish cell-autonomous effects in macrophages. The authors should acknowledge these limitations more explicitly and avoid overextension of the conclusions.

Although n=3 is common in intravital imaging of the meninges, including experiments employing pharmacological manipulations, such as RAMP1 inhibition (5-7), a larger sample size will increase confidence in the results. We further acknowledge that our pharmacological data indicate only a potential role for RAMP1 signaling in meningeal macrophages and that CGRP/RAMP1 signaling in other meningeal immune or vascular cells may also play a role.

**Reviewer #2 (Public review):**
Using chronic intravital two-photon imaging of calcium dynamics in meningeal macrophages in Pf4Cre:TIGRE2.0-GCaMP6 mice, the study identified heterogeneous features of perivascular and non-perivascular meningeal macrophages at steady state and in response to cortical spreading depolarization (CSD). Analyses of calcium dynamics and blood vessels revealed a subpopulation of perivascular meningeal macrophages whose activity is coupled to behaviorally driven diameter fluctuations of their associated vessels. The analyses also investigated synchrony between different macrophage populations and revealed a role for CGRP/RAMP1 signaling in the CSD-induced increase, but not the decrease, in calcium transients.This is a timely study at both the technical and conceptual levels, examining calcium dynamics of meningeal macrophages in vivo. The conclusions are well supported by the findings and will provide an important foundation for future research on immune cell dynamics within the meninges in vivo. The paper is well written and clearly presented.

Thank you.

I have only minor comments.(1) Please indicate the formal definition of perivascular versus non-perivascular macrophages in terms of distance from the blood vessel. This information is not provided in the main text or the Methods. In addition, please explain how the meningeal vasculature was imaged in the main text.

We did not measure the exact distance of the perivascular macrophages from the blood vessels, but defined them as such based on previous data showing that these cells reside along the abluminal surface and maintain tight interactions with mural cells (8). We now provide this information in the revised manuscript, including their labeling approach with a dextran tracer.

(2) Similarly, the method used to induce acute CSD (pin prick) is not described in the main text and is only mentioned in the figure legends and Methods. Additional background on the neurobiology of acute CSD, as well as the resulting brain activity and neuroinflammatory responses, could be helpful.

We have added more background and the method for inducing CSD (i.e., a pinprick in the frontal cortex) in the Results section.

**Reviewer #3 (Public review):**
Strengths:Sophisticated in vivo imaging of meningeal immune cells is employed in the study, which has not been performed previously. A detailed analysis of the distinct calcium dynamics in various subtypes of meningeal macrophages is provided. Functional relevance of the responses is also noted in relation to CSD events.

Thank you for recognizing the strengths of our paper

Weaknesses:(1) The specificity of the methods used to target both meningeal macrophages and RAMP1 is limited. Additional discussion points on the functional relevance of the two subtypes of meningeal macrophages and their calcium responses are warranted. A section on potential pitfalls should be included.

Please see previous responses regarding the specificity of the PF4Cre line for targeting macrophages. The specificity of the RAMP1 antagonist we used (BIBN4096, Olcegepant) has been confirmed by its developer Boehringer Ingelheim, and has been used to target CGRP signaling in numerous studies, including those targeting meningeal macrophage and vascular signaling (2, 7). A section on the study’s limitations has been added.

References:

(1) H. Van Hove et al., A single-cell atlas of mouse brain macrophages reveals unique transcriptional identities shaped by ontogeny and tissue environment. Nat Neurosci 22, 1021-1035 (2019).

(2) F. A. Pinho-Ribeiro et al., Bacteria hijack a meningeal neuroimmune axis to facilitate brain invasion. Nature 615, 472-481 (2023).

(3) G. L. McKinsey et al., A new genetic strategy for targeting microglia in development and disease. Elife 9, (2020).

(4) H. J. Barr et al., The circadian clock regulates scavenging of fluid-borne substrates by brain border-associated macrophages. bioRxiv, (2025).

(5) T. L. Roth et al., Transcranial amelioration of inflammation and cell death after brain injury. Nature 505, 223-228 (2014).

(6) M. V. Russo, L. L. Latour, D. B. McGavern, Distinct myeloid cell subsets promote meningeal remodeling and vascular repair after mild traumatic brain injury. Nat Immunol 19, 442-452 (2018).

(7) K. L. Monaghan et al., Highly dynamic dural sinuses support meningeal immunity. Nature, (2026).

(8) H. Min et al., Mural cells interact with macrophages in the dura mater to regulate CNS immune surveillance. J Exp Med 221, (2024).